# Uncovering Norway: Descriptions of Four New Aphidiinae Species (Hymenoptera, Braconidae) with Identification Key and Notes on Phylogenetic Relationships of the Subgenus *Fovephedrus* Chen

**DOI:** 10.3390/insects15070518

**Published:** 2024-07-10

**Authors:** Korana Kocić, Alf Tore Mjǿs, Jelisaveta Čkrkić, Andjeljko Petrović, Nemanja Popović, Eva Songe Paulsen, Željko Tomanović

**Affiliations:** 1Institute of Zoology, Faculty of Biology, University of Belgrade, Studentski Trg 16, 11000 Belgrade, Serbia; jckrkic@bio.bg.ac.rs (J.Č.); andjeljko@bio.bg.ac.rs (A.P.); nemanja.popovic@bio.bg.ac.rs (N.P.); ztoman@bio.bg.ac.rs (Ž.T.); 2Museum Stavanger, Muségaten 16, 4010 Stavanger, Norway; alf.tore.mjoes@museumstavanger.no (A.T.M.); evasopa@gmail.com (E.S.P.); 3Centre for Biodiversity Genomics, University of Guelph, 50 Stone Road East, Guelph, ON N1G 2W1, Canada; 4Serbian Academy of Sciences and Arts, Knez Mihailova 35, 11000 Belgrade, Serbia

**Keywords:** new species, parasitoids, Norway

## Abstract

**Simple Summary:**

The subfamily Aphidiinae, comprising over 500 known species globally, consists of obligatory endoparasitoids of aphids. Recognised for their role as biological control agents of various aphid pests, this group is considered relatively well-studied, particularly in Europe. However, Norway’s documented Aphidiinae diversity is notably low, with only 33 reported species. Our study aimed to assess the true biodiversity of aphid parasitoids in Norway using malaise traps. The results of morphological and molecular analyses identified several specimens distinct from any known species. In this paper, we describe four species new to science across three genera, provide an identification key for species of the subgenus *Fovephedrus*, and discuss their phylogenetic relationships.

**Abstract:**

With only 33 reported species, Norway ranks among the European countries with the lowest documented diversity of parasitoids from the subfamily Aphidiinae. The “MUST Malaise” project, carried out by Museum Stavanger in Norway, aimed to assess insect abundance and biodiversity and create a reference base for future studies. The preliminary results of our study revealed four species new to science, indicating that the current number of recorded species in Norway is significantly lower than the actual diversity. All species possess unique combinations of morphological characters, distinguishing them from other known Aphidiinae species. Molecular analysis of the barcoding region confirmed that these specimens all belong to the previously undescribed species. In this study, we describe *Aphidius norvegicus* sp.n., *Praon breviantennalis* sp.n., *Ephedrus gardenforsi* sp.n., and *Ephedrus borealis* sp.n., all collected in Norway. We also provide an identification key and discuss the phylogenetic relationships within the subgenus *Fovephedrus* Chen, 1986.

## 1. Introduction

Recent reports on insect biodiversity and overall biomass numbers have drawn attention over the past decades, highlighting the consequences if this declining trend continues. Anthropogenic influence through agricultural intensification (i.e., habitat loss and use of pesticides) and climate change are now recognised as the main drivers of biodiversity loss. It is estimated that around 40% of the world’s insect species are threatened with extinction within the next several decades [1]. As this study reported, in addition to Lepidoptera and some Coleoptera (dung beetles), Hymenoptera are among the most affected terrestrial insect groups [1]. Furthermore, the global biodiversity loss is accompanied by a decline in insect biomass. Hallmann et al. [2] reported a staggering 76% reduction in insect biomass in several protected reserves in Germany. Due to these alarming projections, there is a growing intent to record the diversity of insect species globally.

In Europe, the subfamily Aphidiinae, comprising koinobiont endoparasitoids of aphids, is considered a relatively well-studied group. Extensive research efforts by experts have resulted in a high number of reported species in some regions, especially central, western, and southeastern Europe (Czech Republic, France, Serbia, Montenegro, Greece). For comparison, there are currently 121 recorded Aphidiinae species in Serbia, 135 in the Czech Republic, and 85 in Montenegro [3]. On the other hand, some areas in Europe remained unexplored, including Norway, which has only 33 recorded species [4]. This number is surprisingly low, especially considering that 119 species are recorded in neighbouring Sweden [5]. The “MUST Malaise” project, carried out by Museum Stavanger in Norway, aimed to assess insect abundance and the biodiversity of different insect groups to create a reference base for future studies. Preliminary analysis of the collected material revealed that the species richness of the subfamily Aphidiinae in Norway is high and comparable to that of other European countries (unpublished data). Among this material, multiple specimens from three different genera exhibited unique combinations of morphological characters, distinguishing them from other known Aphidiinae species. Molecular analysis of the cytochrome oxidase I (COI) barcoding region confirmed that these specimens all belong to previously undescribed species. In this manuscript, we describe four species new to science, collected in Norway. We also provide an identification key and discuss the phylogenetic relationships within the subgenus *Fovephedrus* Chen, 1986.

## 2. Material and Methods

### Sampling and Morphological Analysis

The material was collected during 2020–2022 using malaise traps in Norway (Figure 1), from different localities in Rogaland county (Table 1). One additional specimen was collected as an aphid mummy in Northern Ireland, UK. In both cases, specimens were transferred to and stored in 96% ethanol. 

The material was identified to the genus level. Three females were designated as *Aphidius* sp., ten as *Praon* sp., 20 (males and females) as *Ephedrus* sp. 1 and one as *Ephedrus* sp. 2. Specimens were dissected and slide mounted in Berlese medium. Photographs of the dissected specimens were taken with a Leica DM LS phase contrast microscope (Leica Microsystems GmbH, Wetzlar, Germany). The obtained photographs were stacked in Helicon Focus software (version 7.6.1; HeliconSoft, Kharkiv, Ukraine). ImageJ software [6] was used to measure all important taxonomical characters. Morphological terminology follows Sharkey and Wharton [7]. All material is deposited in the collection of the Institute of Zoology, Faculty of Biology, University of Belgrade (Serbia).

After the initial examination under the ZEISS Discovery V8 stereomicroscope (Carl Zeiss MicroImaging GmbH, Göttingen, Germany) and prior to slide mounting, the non-destructive extraction of the total DNA was performed with the Qiagen DNeasy^®^ Blood & Tissue Kit (Qiagen Inc., Valencia, CA, USA). Universal primers LCO1490 and HCO2198 [8] were used for PCR amplification of the barcode region (COI). The final volume of the amplification mixture (50 µL) contained 32.6 µL of nuclease-free water, 10 µL of buffer, 1 µL of each primer pair, 1 µL of nucleotides, 0.4 µL of polymerase, and finally 4 µL of extracted DNA. The following PCR temperature profile was used: 60 s of initial denaturation, 35 cycles of 60 s denaturation (94 °C), 60 s annealing (54 °C), 90 s extension (72 °C), and 7 min of final extension (72 °C). Amplification products were purified and sequenced by Macrogen Europe (Netherlands). Obtained sequences were visualised, trimmed and aligned in BioEdit software [9]. The analysis of evolutionary divergence was conducted using MEGA 6 [10] software using the Kimura 2-parameter distance model [11]. In all molecular analyses, the reconstruction of phylogenetic relationships was conducted with the Maximum Likelihood method. The Tamura–Nei [12] parameter model with discrete Gama distribution was proposed as the best-fitting model and was conducted with 2000 bootstrap replications. Seven sequences from the *Aphidius urticae* sensu stricto group were acquired from GenBank, as well as one sequence of *Ephedrus persicae* Froggatt (outgroup). For the molecular analysis of *Praon* sp. sequences, seven sequences of the *Praon abjectum* species group were acquired from the GenBank as well as one sequence of *Ichneumon picticollis* Holmgren, 1864. An additional 13 sequences of the subgenus *Fovephedrus* were acquired from GenBank. As the outgroup, six sequences of the subgenus *Ephedrus* were used, as well as one sequence of *I. picticollis*.

## 3. Results

The three COI sequences recovered from specimens designated as *Aphidius* sp. separated from other *Aphidius* sequences with the mean between species evolutionary divergence ranging from 6.8% to 11.1%. Within the *Aphidius urticae* s. str. group, the mean between species genetic distances ranged from 8.0% to 9.7% (Figure 2). With three sequences representing three different haplotypes, the intraspecific genetic distance was 0.3–0.7% (GenBank accession numbers PP856231–PP856233). Seven sequences of the specimens designated as *Praon* sp. were separated from other sequences of the genus *Praon* with genetic distances ranging from 5.9% to 9.8%, representing a single haplotype (GenBank accession numbers PP856248–PP856254). On the phylogenetic tree of the *Praon abjectum* species group, these sequences clustered separately with the mean between species genetic distance ranging from 5.9% (*P. longicaudus*) to 9.5% (*P. sambuci*) (Figure 3). A total of 14 recovered sequences of the specimens designated as *Ephedrus* sp. 1 showed no genetic variability, all belonging to the single haplotype (GenBank accession numbers PP856234–PP856247). In the phylogenetic tree, all *Ephedrus* sp. 1 sequences clustered separately within subgenus *Fovephedrus* (*Ephedrus persicae* species group) with genetic distances ranging from 5.1% (*Ephedrus chaitophori* Gärdenfors, 1986) to 6.3% (*Ephedrus persicae* Froggatt, 1904). A single specimen sequence, designated as *Ephedrus* sp. 2 (GenBank accession number PP856255), also grouped within the subgenus *Fovephedrus*, with an evolutionary divergence of 5.1%–6.4% compared to other members of the subgenus (Figure 4). 

The analysis of the morphological characters also revealed that all four species (*Aphidius* sp., *Praon* sp., *Ephedrus* sp. 1 and *Ephedrus* sp. 2) possess unique morphological traits or a combination of morphological characteristics that clearly distinguish them from other congeners.

Based on molecular and morphological evidence, we describe four species new to science, all collected from Norway: *Aphidius norvegicus* sp.n., *Praon breviantennalis* sp.n., *Ephedrus borealis* sp.n. and *Ephedrus gardenforsi* sp.n.

### 3.1. New Species Descriptions

#### 3.1.1. *Aphidius norvegicus* sp.n. Tomanović & Kocić (Figure 5) 


https://zoobank.org/6E5CD806-B0F2-486D-BFAA-F3744C1CE992


(GenBank Accession Numbers PP856231–PP856233)

*Diagnosis*. Morphologically, *A. norvegicus* sp.n. resembles species from *Aphidius urticae* s.str. group (*Aphidius urticae* Haliday, 1834, *Aphidius rubi* Starý, 1962, *Aphidius silvaticus* Starý, 1962) with respect to the costulated anterolateral area of the petiole, number of flagellomeres, proportion of length and width of flagellomere 1 and tergite 1. However, it can easily be discriminated by more elongated pterostigma (proportion of length and width of pterostigma 4.30–4.50 in *A. norvegicus* sp.n., while in the *A. urticae* s.str. group it is 3.40–4.10). The R1 vein (=metacarpus) in *A. norvegicus* sp.n. is shorter than pterostigma (the proportion between the pterostigma and R1 length is 1.20–1.50) while in *A. urticae* and *A. silvaticus,* pterostigma is subequal to the R1 vein. Additionally, *A. norvegicus* sp.n. has a length-to-width ratio of flagellomere 2 of 3.40–3.60, compared to 2.70–3.30 in *A. rubi.*

*Description*. **Female**. **Head**. Eyes oval, sparsely setose (Figure 5A). Malar index 0.16–0.18 times the longitudinal eye diameter. Clypeus oval, with 15–22 long setae (Figure 5A). Tentorial index (tentoriocular line/intertentorial line) 0.33–0.37. Antennae with 18–19 flagellomeres, with semierect setae which are subequal to half the segment diameter (Figure 5B). Flagellomere 1 (F1) and flagellomere 2 (Figure 5C) are 3.50–3.80 and 3.40–3.60 times as long as the median width, respectively. F1 is subequal to F2. F1 without and F2 with 1–4 longitudinal placodes (Figure 5C). Maxillary palps with four palpomeres, labial palps with three palpomeres. **Mesosoma**. Mesonotum with notaulices distinct only at the ascending part, slightly crenulated, with two rows of setae (Figure 5D). Propodeum areolated with narrow central pentagonal areola (Figure 5E). External and dentiparal areolae with 4–10 and 2–7 setae, respectively (Figure 5E). **Fore wing**. Pterostigma elongated, approx. 4.30–4.50 times as long as wide and approx. 1.20–1.50 times as long as the distal abscissa of R1 (=metacarpus) (Figure 5H); R1 vein 3.00 times longer than pterostigma width. **Metasoma**. Petiole 3.30–3.50 times as long as wide at the spiracle level (Figure 5F), with a prominent and adjective mediodorsal carina, with 7–9 costulae on its anterolateral area and about 10 setae in the lower dorsal part. The ovipositor sheath is almost straight in its dorsal margin (Figure 5G). 

*Colour*. Body generally yellow to light brown. Head yellow to light brown, eyes dark brown. Scape yellow, pedicel light brown to brown, with a narrow yellow ring at the base of flagellomere 1. The remaining part of the antennae is brown. Legs yellow with dark apices. The remainder of the body is light brown.

*Body length*. 1.7–2.1 mm

**Male.** Unknown

*Aphid host*. Tribe Macrosiphini.

*Distribution*: Norway, Northern Ireland.

*Etymology*. The new species was named after the locus typicus of the type specimen.

Material. Norway: **holotype** ♀, Rogaland, Suldal, Vårvik (59.54282N, 6.64587E), 27.06.2021–03.08. 2021, leg. Eva Songe Paulsen and Rune Roalkvam, collected by malaise trap; **paratypes**: 1♀, Rogaland, Sandnes, Svanholmen (58.88880N, 5.70937E), 01.06.2022–15.06.2022, malaise trap, leg. Alf Tore Mjøs. United Kingdom, Northern Ireland: 1♀, Moneyreagh (54.532919N, 5.849970E), 2022, leg. Martin Wohlfarter (collaborative collection efforts of Koppert B.V. (Martin Wohlfarter) and Koppert UK Ltd. (Jasper Hubert and David Davidson), in the organic vegetable farm with the diverse adjacent garden, sampled aphid mummy from the *Rosa* sp. Holotype and paratypes slide mounted and deposited in the collection of the Institute of Zoology, Faculty of Biology, University of Belgrade, Serbia.

**Figure 5 insects-15-00518-f005:**
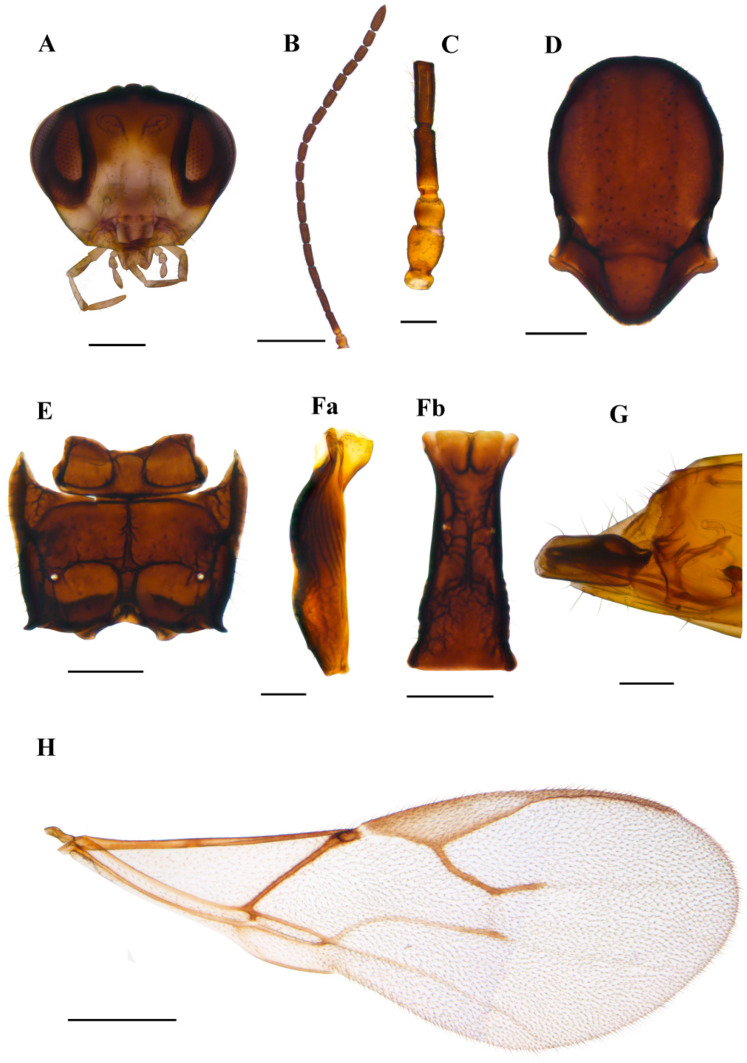
*Aphidius norvegicus* sp. n. holotype female (**A**) head (**B**) antenna (**C**) scape, pedicel, first and second flagellomere (**D**) mesonotum—dorsal aspect (**E**) propodeum—dorsal aspect (**Fa**) petiole—lateral aspect (**Fb**) petiole—dorsal aspect (**G**) ovipositor sheath—lateral aspect (**H**) fore wing. Scale bars: 200 µm (**A**,**D**,**E**); 500 µm (**B**,**H**); 100 µm (**C**,**F**,**G**).

#### 3.1.2. *Praon breviantennalis* sp.n. Kocić & Tomanović (Figure 6)


https://zoobank.org/8C159F79-3C7C-4E0D-AF49-C147F85C6A92


(GenBank Accession Numbers PP856248–PP856254)

*Diagnosis*. *Praon breviantennalis* sp.n. can easily be discriminated from other *Praon* species by short antennae (13–14 antennomeres) and a very long R1 vein, which is equal in length to the pterostigma.

*Description*. **Female**. **Head**. Malar space equal to 0.10–0.15 of longitudinal eye diameter. Clypeus oval, with 10–16 long setae. Tentorial index 0.20–0.28 (Figure 6A). Maxillary palps with four palpomeres, labial palps with three palpomeres. Antennae with 13–14 antennomeres, moderately thickened, with semierect setae, which are subequal to or shorter that half of the segment diameter (Figure 6B). F1 and F2 are moderately elongated, 3.50–4.00 and 2.70–3.20 times as long as wide, respectively (Figure 6C). F1 is 1.30–1.40 times longer than F2. F1 without, and F2 without or with 1 (2) longitudinal placodes. **Mesosoma**. Mesonotum (Figure 6D) with central lobe covered with dense setae. Lateral lobes of mesonotum with large glabrous areas. Notaulices deep and distinct throughout. Propodeum (Figure 6E) smooth, densely pubescent with small central glabrous areas. **Forewing**. Pterostigma 3.40–3.70 times as long as wide (Figure 6H) and equal to distal abscissa of R1 (=metacarpus). The first half of the m–cu vein is usually sclerotized and coloured; the remaining part is colourless; Rs+M vein colourless (Figure 6H). **Metasoma**. Petiole elongated, 1.10–1.40 times as long as wide (Figure 6F) at the level of the spiracles. The distance between spiracles and the apex is less than the width at the level of spiracles (proportion between width at the level of spiracles and distance between spiracles and apex of petiole 1.75–2.10). Petiole with 2–4 long setae along the petiole sides. Ovipositor sheath (Figure 6G) with a slightly concave dorsal margin. Apex round, with two conical spines on its upper and lower edge. 

*Colour*. Head brown, with mouthparts yellow to light brown. Scape and pedicel brown, annellus yellow, 1/3–2/3 of flagellomere 1 yellow or light brown. The remaining parts of the antennae are brown. Mesosoma and metasoma are light brown to brown. Legs yellow with dark apices. The petiole is yellow to light brown. 

*Body length*. 1.4–1.7 mm

**Male.** Unknown

*Aphid host*: Unknown.

Distribution: Norway.

*Etymology*. The new species was named after the unique character among the congeners, short antennae. 

Material. **Holotype**: 1♀ Norway, Rogaland, Sandnes, Svanholmen (58.88878N, 5.71121E), 15.08.2021–31.08.2021, leg. Alf Tore Mjǿs, malaise trap **paratypes**: 1♀, same as holotype; 2♀ same as previous, 16.07.2021–31.07.2021, 2♀ same as previous, 31.08.2021–15.09.2021; 1♀ Rogaland, Sola, Roynebergsletta plen (58.90041N, 5.68971E), 15.06.2021–29.06.2021, leg. Alf Tore Mjǿs, malaise trap; 2♀ same as previous 15.09.2021–01.10.2021, 1♀ same as previous 29.06.2021–16.08.2021. Holotype and eight paratypes slide mounted, one female stored in ethanol. Material is deposited in the collection of the Institute of Zoology, Faculty of Biology, University of Belgrade, Serbia.

**Figure 6 insects-15-00518-f006:**
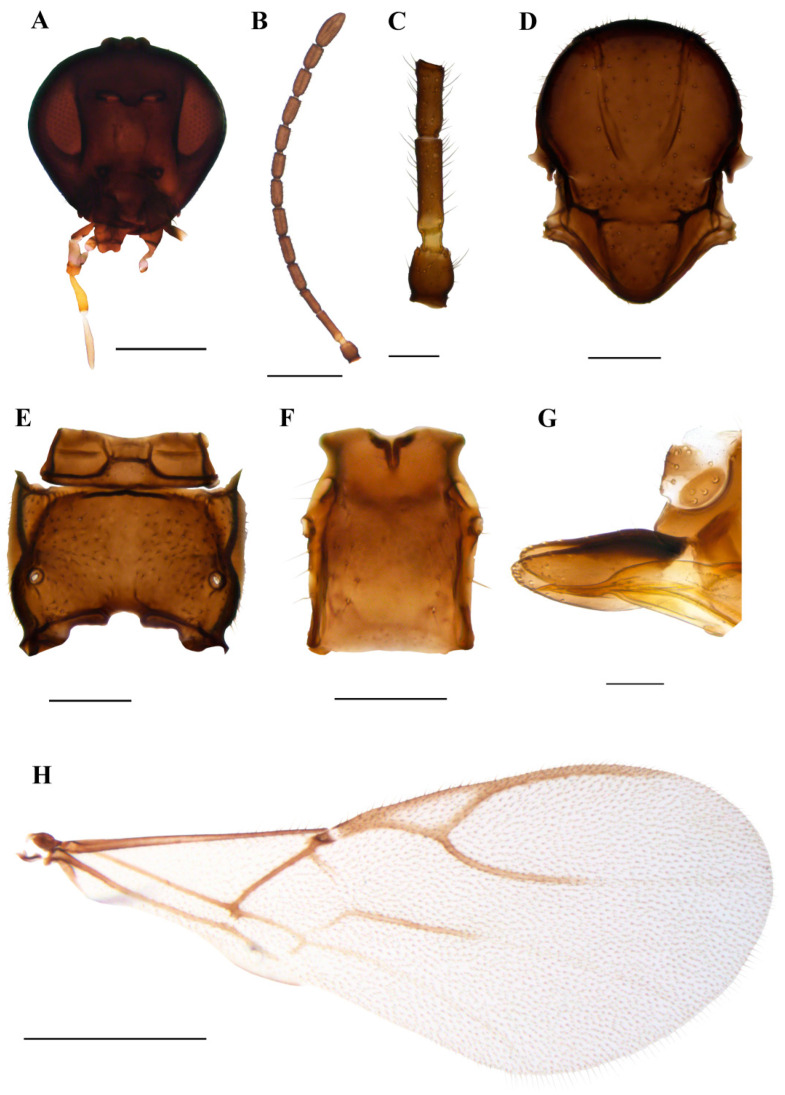
*Praon breviantennalis* sp. n. holotype female (**A**) head (**B**) antenna (**C**) scape, pedicel, first and second flagellomere (**D**) mesonotum—dorsal aspect (**E**) propodeum—dorsal aspect (**F**) petiole—dorsal aspect (**G**) ovipositor sheath—lateral aspect (**H**) fore wing. Scale bars: 200 µm (**A**,**D**,**E**); 500 µm (**B**,**H**); 100 µm (**C**,**F**,**G**).

#### 3.1.3. *Ephedrus gardenforsi* sp. n. Kocić & Tomanović (Figure 7)


https://zoobank.org/53AA9E93-BA40-4795-AA47-5F0037596AC0


(GenBank Accession Number PP856255)

*Diagnosis*. The new species belongs to the *Fovephedrus* subgenus (former *Ephedrus persicae* species group), based on the short fore wing 3RSa vein. It differs from the nominative species *E. persicae* by the longer F1, which is lightly coloured (F1 l/w is 4.8 in *E. gardenforsi* vs 3.8–4.2 in *E. persicae*). It differs from *Ephedrus lonicerae* Tomanović, Kavallieratos & Starý, 2009 with respect to the length and colouration of F1 as well as the number of foveal pits on mesoscutum (in *E. lonicerae* F1 is 5.0–5.3 times as long as wide, F1 and half of F2 are yellow, the mesoscutum has two foveal pits, while in *E. gardenforsi* sp.n., F1 is about 4.8 times as long as wide, F1 is yellowish at the first part and the mesoscutum has one foveal pit). It differs from *Ephedrus tamaricis* Tomanović & Petrović, 2016 with respect to the colouration of F1 (yellowish at the first part), a shorter pterostigma (pterostigma 4.8–5.0 times as long as wide) and a more elongated petiole (1.5 times as long as wide), while in *E. tamaricis,* F1 has only a basal yellow ring with the remainder of the flagellomere being brown. The pterostigma length/width ratio is 5.0–5.22, and the petiole is 1.2–1.4 times as long as wide at the spiracle level. The new species is most similar to *E. chaitophori*; however, in this species, F8 and F9 are visibly separated, thus not forming a club as in *E. gardenforsi* (Figure 7B). The ovipositor sheath in *E. gardenforsi* lacks a subapical constriction, which is present in *E. chaitophori.*

*Description*. **Female**. **Head**. Eyes medium-sized, without noticeable setae (Figure 7A). Clypeus is slightly convex, bearing sparse long setae. Tentorial index 0.5, tentorial pits wide and deep. Malar space equal to 0.35 of longitudinal eye diameter. Mandibles bidentate. Maxillary palps with four palpomeres, labial palps with two palpomeres, both covered with sparse long setae. Antennae with 11 antennomeres with long setae that are subequal to segment diameter. The last two apical segments (F8 and F9) are not well separated, giving the impression of a club (Figure 7B). The first flagellar segment (F1) is 1.44 times as long as the second flagellar segment (F2) (Figure 7C). F1 is elongated, 4.8 times as long as wide, bearing 1–2 longitudinal placodes (=rhinariae). The second flagellar segment (F2), with 2–3 longitudinal placodes, is 2.9 times as long as wide (Figure 7C). **Mesosoma**. Mesoscutum with notaulices present only at the ascending part (Figure 7D). The mesoscuteal fovea is present, yet small and feebly visible. Distinctive setae are present on the surface of the mesoscutum, following the direction of notaulices and evenly distributed around the foveal pit. Propodeum areolated, with 4–5 setae present on external areolae and smooth dentiparal areolae (Figure 7E). **Fore wing** pterostigma 4.8–5.0 times as long as wide. 3RSa/2RS and 3RSb/3Rsa vein ratios 0.76–0.78 and 3.0–3.3, respectively. 3RSa vein 1.2 times longer than r-m (Figure 7H). **Metasoma**. Petiole short, 1.5 times as long as wide, bearing rugosities along the dorsal surface and with 5 long setae on lateral sides (Figure 7F). The ovipositor sheath is elongated, 3.1 times as long as wide, with 4 and 3 setae along the dorsal and ventral side, respectively (Figure 7G).

*Colour*. Head brown, mouthparts light brown. Scape and pedicel brown, anellus yellow, F1 yellowish in the first part, gradually getting brown towards the distal part. The remainder of the flagellar segments are brown. The mesonotum, propodeum, petiole, abdomen and ovipositor sheath are brown. Legs yellowish. Fore wings hyaline.

*Body length*. 1.5 mm.

**Male**: unknown

Aphid host: unknown

Distribution: Norway.

*Etymology*: The new species is named in honour of the Swedish entomologist Ulf Gärdenfors, who has greatly contributed to the knowledge of the subfamily Aphidiinae, especially the genus *Ephedrus*.

Material: **Holotype**: 1♀, Norway, Sola, Forus, Røynebergsletta, (58.90017N, 5.68956E), 29.06.2021–16.07.2021, collected using a malaise trap, leg. Alf Tore Mjøs. Holotype slide mounted and deposited in the collection of the Institute of Zoology, Faculty of Biology, University of Belgrade.

**Figure 7 insects-15-00518-f007:**
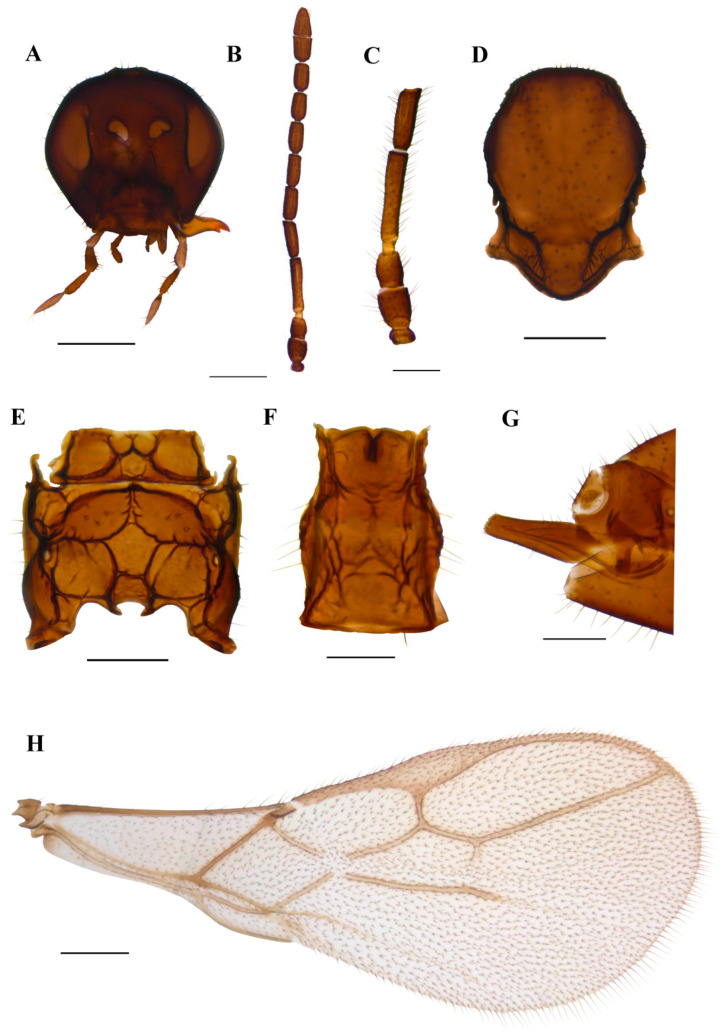
*Ephedrus gardenforsi* sp. n. holotype female (**A**) head (**B**) antenna (**C**) scape, pedicel, first and second flagellomere (**D**) mesonotum—dorsal aspect (**E**) propodeum—dorsal aspect (**F**) petiole—dorsal aspect (**G**) ovipositor sheath—lateral aspect (**H**) fore wing. Scale bars: 200 µm (**A**,**D**,**E**); 500 µm (**B**,**H**); 100 µm (**C**,**F**,**G**).

#### 3.1.4. *Ephedrus borealis* sp.n. Kocić & Tomanović (Figure 8)


https://zoobank.org/AE14A00F-E263-498B-BF72-C491FEDFE492


(GenBank Accession Numbers PP856234–PP856247)

*Diagnosis.* The new species belongs to the subgenus *Fovephedrus* (*Ephedrus persicae* species group) due to the ratio of 3RSa/2RS. It differs from all other species of the subgenus with respect to a very long first flagellar segment F1 (5.7–6.3), elongated petiole (1.9–2.3) and ovipositor sheath, a narrow central areola of propodeum and a densely pubescent body. 

*Description*. **Female**. **Head**. Eyes medium sized, head with scattered long setae along the surface. Clypeus convex, bearing 10 long setae. Tentorial index 0.4–0.5, tentorial pits wide. Malar space equal to 0.19–0.22 of longitudinal eye diameter. Maxillary palps with four palpomeres, labial palps with two palpomeres, both covered with multiple long setae (Figure 8A). Antennae with 11 antennomeres with long setae that are slightly longer than the segment diameter (Figure 8B). The last two apical segments (F8 and F9) are well separated. The first flagellar segment (F1) is 1.37–1.55 times as long as the second flagellar segment (F2) (Figure 8C). F1 is long, 5.7–6.3 times as long as wide, without longitudinal placodes (=rhinariae). The second flagellar segment (F2) usually has one longitudinal placode, sometimes has none, and is 3.0–3.5 times as long as wide. **Mesosoma**. A mesoscutum with notaulices is present only at the ascending part, with two longitudinal rows of setae. In most specimens, one mesoscuteal fovea is clearly visible (Figure 8Da). In several, two foveal pits are visible, irregular, elongated and shallow (Figure 8Db). Propodeum with narrow central areola, with 3–5 setae present on external and 0–1 on dentiparal areolae (Figure 8E). **Fore wing** pterostigma 5.5–6.0 times as long as wide. 3RSa/2RS and 3RSb/3Rsa vein ratios 0.6–0.75 and 3.2–3.5, respectively. 3RSa vein 1.1–1.2 times longer than r-m (Figure 8H). **Metasoma**. Petiole 1.9–2.3 times as long as wide, bearing rugosities along the dorsal surface and with 3 long setae on lateral sides (Figure 8F). The ovipositor sheath is moderately elongated, with 2 and 1 setae along the dorsal and ventral sides, respectively (Figure 8G).

*Colour*. Head brown, mouthparts yellowish. Scape brown to light brown, pedicel and annelus yellow, 2/3 or complete F1 yellow. Sometimes F2 has a yellow or light brown basal ring, grading into darker brown at the distal part. The remainder of the flagellar segments are brown. The mesonotum, propodeum, petiole and abdomen are yellowish brown to brown. The ovipositor sheath is brown. Legs yellow. Fore wings hyaline. 

*Body length*. 1.5–1.7 mm.

**Male**. Eyes smaller than in females. The first flagellar segment is 1.15–1.20 times as long as the second flagellar segment, bearing 0–1 and 1–2 longitudinal placodes, respectively. F1 and F2 are 4.3–4.4 and 2.7–2.8 times as long as wide. The mesonotum has notaulices only at the ascending part, with two longitudinal rows of setae. The mesoscuteal foveal pit is present, deep and narrow, or the mesoscutum has two connected, shallow and irregular pits. The propodeum is as in females, with narrow central pentagonal areola and 2 and 1 setae present on the external and dentiparal areolae, respectively. Fore wing pterostigma is 5.5 times as long as wide, 3RSa/2RS veins ratio 0.6–0.65. Petiole is 1.7–1.8 times as long as wide at the spiracle level. Male genitalia as in Figure 6I. The colour is slightly darker than in females. Scape and pedicel are brown, annelus is light brown or yellow. F1 is yellow or light brown in the first 2/3 of the segment, and the remaining flagellar segments are brown.

*Aphid host*. Unknown.

Distribution: Norway.

*Etymology*: Name of the new species derived from its known distribution, *borealis* meaning northern in Latin.

Material: **Holotype** ♀ Norway, Rogaland, Hå, Brusand (58.53837N, 5.74777E), 31.08.2020–17.09.2020, malaise trap, leg. Alf Tore Mjǿs **paratypes**: 2♂, same as holotype; 1♀1♂ Rogaland, Time, Linemyra (58.71556N, 5.63867E), 31.08.2021–30.09.2021, malaise trap, leg. Alf Tore Mjøs, 1♀1♂ same as previous, 01.08.2021–30.08.2021, 2♀4♂ same as previous, 29.06.2021–20.07.2021, 3♀ same as previous, 21.07.2021–31.08.2021, 2♀2♂ same as previous, 01.06.2021–29.06.2021.

Holotype and seven female and six male paratypes slide mounted, two female and four male paratypes stored in ethanol. Material is deposited in the collection of the Institute of Zoology, Faculty of Biology, University of Belgrade, Serbia.

**Figure 8 insects-15-00518-f008:**
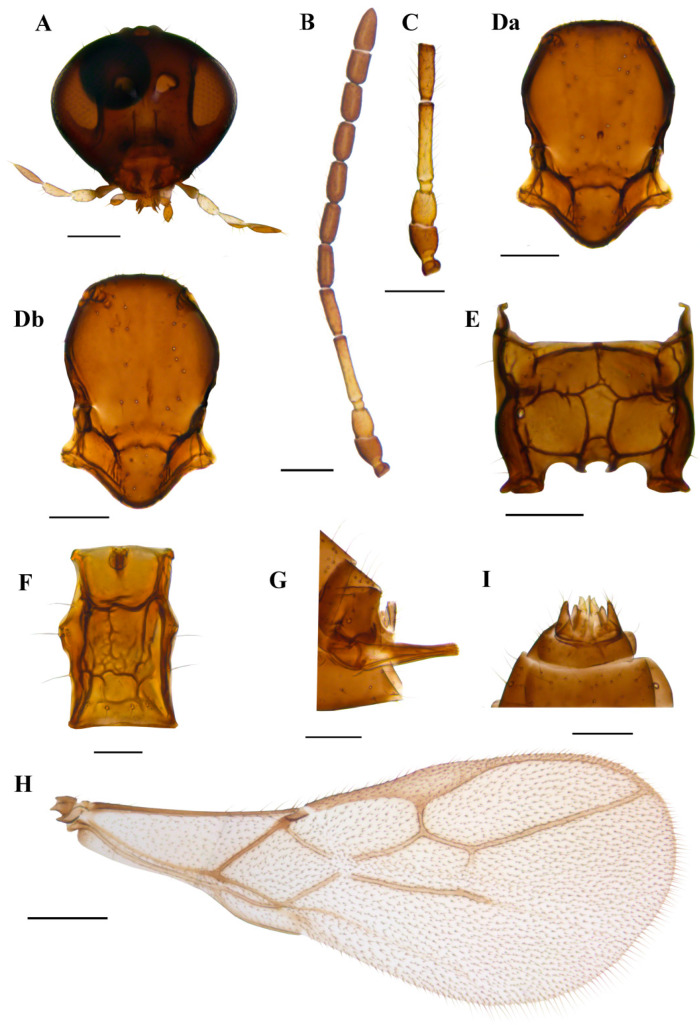
*Ephedrus borealis* sp. n. holotype female (**A**) head (**B**) antenna (**C**) scape, pedicel, first and second flagellomere (**Da**) mesonotum—dorsal aspect (**E**) propodeum—dorsal aspect (**F**) petiole—dorsal aspect (**G**) ovipositor sheath—lateral aspect (**H**) fore wing; paratype female (**Db**) mesonotum—dorsal aspect; paratype male (**I**) aedeagus. Scale bars: 200 µm (**A**,**D**,**E**); 500 µm (**B**,**H**); 100 µm (**C**,**F**,**G**).


Key for Identification of Female Parasitoids Belonging to Subgenus *Fovephedrus* (*Ephedrus persicae* Species Group)



1Number of antennomeres 12, central areola of propodeum not clearly defined......................................………………*E. antennalis* Tomanović
–Number of antennomeres 11, central areola of propodeum clearly defined………………………………………………………………………...2
2F1 very long, 5.7–6.3 times as long as wide, narrow central propodeal areola, petiole elongated, 1.9–2.3 times as long as wide at spiracle level (Figure 8) …………………………………………………………………………………………......……….*E. borealis* sp.n. Kocić & Tomanović
–F1 3.8–5.3 times as long as wide, wide central areola of propodeum, petiole subquadrate, 1.1–1.6 times as long as wide at spiracle level…………………………………………………………………………………………………………………………………………………………....3
3Mesoscutum with two foveal pits, F1 elongated, 5.0–5.3 times as long as wide.........……….....*E. lonicerae* Tomanović, Kavallieratos & Starý
–Mesoscutum with one foveal pit, F1 3.8–4.9 times as long as wide …………………………………………………………………………………...4
4F1 uniformly brown, 3.8–4.2 times as long as wide………………………………………………………………………………..*E. persicae* Froggatt
–F1 with the yellow ring at the base, or 1/2 to 2/3 yellow, 4.5–4.9 times as long as wide………………………………………………………….....5
5F1 with narrow yellow ring at the base, F1 subequal to F2 (1.2–1.3)…….…………………..…………………*E. tamaricis* Tomanović & Petrović
–Approx. 1/2 to 2/3 of F1 yellow, F1 1.3–1.5 times longer than F2…………………………................………………………………………………...6
6Flagellomeres F8 and F9 well separated, ovipositor sheath elongated with subapical constriction……………….....*E. chaitophori* Gärdenfors
–F8 and F9 not well separated, forming a club, ovipositor sheath without subapical constriction (Figure 7)..…………………………………………………………………………………………………......................*E. gardenforsi* sp.n. Kocić & Tomanović


## 4. Discussion

Only sporadic faunistic research articles on Braconidae have reported data on the species found in Norway [13,14,15,16]. The preliminary results of our study revealed four species new to science, indicating that the current number of recorded species in Norway is significantly lower than the actual diversity. All specimens, except for one female reared from an aphid mummy, were collected using malaise traps. Although this type of sampling results in the loss of data on tritrophic interactions (plant–aphid–parasitoid), the diversity of the sampled material often greatly surpasses the number of species collected by rearing. 

The genus *Aphidius*, with more than 130 described species [17] distributed worldwide, is one of the largest members of the subfamily Aphidiinae. Several species, including *Aphidius colemani* Viereck, *A. ervi* Haliday, and *A. matricariae* Haliday are used in biological control programs for pest aphids and are commercially distributed across the world. Only 11 species of this genus have been reported from Norway [4]. A newly identified species, *Aphidius norvegicus* sp. n., morphologically belongs to the *A. urticae* sensu stricto group, which includes *A. urticae*, *A. rubi* and *A. silvaticus* [18,19]. Phylogenetic reconstruction of the barcoding region was in congruence with morphological analysis, revealing that the new species is genetically distinct from other species of the group, exhibiting significantly high evolutionary divergence (Figure 2). Although the tritrophic associations of the material collected from Norway are currently unknown, an additional female was reared from an aphid mummy of the tribe Macrosiphini found on *Rosa* sp. Three additional sequences of the same species are available in the BOLD database (GMNWJ1781-14 GMNWK3447-14, GMNWK2015-14), all originating from Norway. 

Currently, there are 33 recorded species of the genus *Praon* in Europe, with only seven reported from Norway [4]. The host range of these species varies from strict specialists to generalists like *Praon volucre* (Haliday, 1833), which parasitizes a broad range of aphid hosts. The species within this genus exhibit variability in the number of antennal segments, ranging from the lowest documented in *Praon necans* Mackauer, 1959 (15–16), *Praon abjectum* Haliday, 1833 (15–16) *Praon rosaecola* Starý, 1961 and *Praon staryi* Kavallieratos & Lykouressis, 2000 (usually 16–17 in both species) to a maximum of 23 in *Praon longicorne* Marshall, 1896. The newly described *P. breviantennalis* sp.n. possesses a unique character among its congeners: very short antennae consisting of only 13–14 antennomeres. Based on morphological analysis results, this species belongs to the *Praon abjectum* species group [18,20]. In the phylogenetic tree, sequences belonging to *P. breviantennalis* sp. n. formed a separate clade and confirmed that this species is genetically distinct from other taxa (Figure 3). 

The genus *Ephedrus*, considered basal within the subfamily Aphidiinae and containing several important biological control agent species, has been relatively well–studied, particularly in Europe. Out of more than 40 described species, only four have been detected in Norway, all belonging to the subgenus *Ephedrus* Haliday [4]. The first comprehensive review of the Palearctic species of the genus was provided by Gärdenfors [21], where *E. persicae* species group was established with two members, *E. persicae* and the newly described *E. chaitophori* Gärdenfors. Over the past decades, additional new species belonging to the *E. persicae* group have been described—*E. lonicerae* Tomanović, Kavallieratos & Starý [22], *E. tamaricis* Tomanović & Petrović [23], and *E. antennalis* Tomanović [24]. Kocić et al. [25] proposed a new subgeneric classification and raised the species from the *E. persicae* species group to the subgenus *Fovephedrus* Chen, 1986. Based on morphological and molecular analyses, two new species (*E. borealis* sp.n. and *E. gardenforsi* sp.n.) both belong to *Fovephedrus* (*Ephedrus persicae* species group) and are the first reports of this subgenus in Norway. *Ephedrus borealis* sp. n. exhibits clear morphological characters that easily distinguish it from other taxa, such as a long first flagellar segment, a narrow pentagonal areola of propodeum (unusual for the genus), and a densely setose body. Although its aphid host is unknown, it is possible that this species parasitizes underground aphids or other hosts that produce a large amount of honeydew, with the dense setae across its body surface serving as an adaptation to the special conditions it encounters while searching for a host. Another new member of the genus, *Ephedrus gardenforsi* sp. n., has so far been collected only once. This species morphologically resembles *E. chaitophori*, and it is possible that it has been misidentified in previous samples. Therefore, a careful re-examination of the *E. chaitophori* material should be conducted. Prior to this study, only five species of the subgenus were known (*E. persicae*, *E. tamaricis*, *E. chaitophori, E. antennalis* and *E. lonicerae*). The results of our molecular analysis position both *E. borealis* and *E. gardenforsi* close to *E. chaitophori*, a specialized parasitoid of aphids found on poplars (*Populus*). 

Tritrophic interactions are crucial for studying Aphidiinae, as they provide key insights into their biology, host specificity, and distribution, which align with those of their aphid hosts. In addition to continuing the exploration of the Norwegian Aphidiinae fauna through malaise trapping, future studies should focus on rearing from aphid hosts to reveal the tritrophic interactions of the newly described species.

## Figures and Tables

**Figure 1 insects-15-00518-f001:**
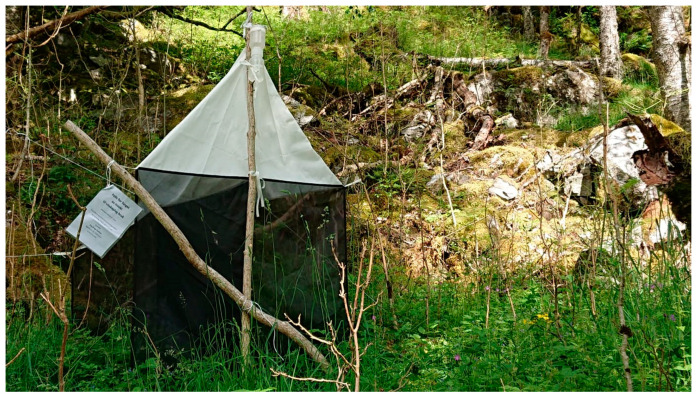
Malaise trap Vårvik, Suldal, Rogaland, Norway, 2021. Photo: Rune Roalkvam.

**Figure 2 insects-15-00518-f002:**
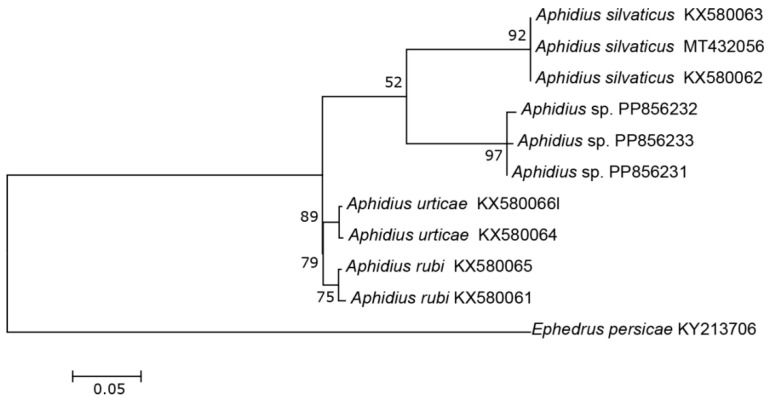
The evolutionary history of the *A. urticae sensu stricto* group, inferred by using the Maximum Likelihood method based on the Tamura–Nei model.

**Figure 3 insects-15-00518-f003:**
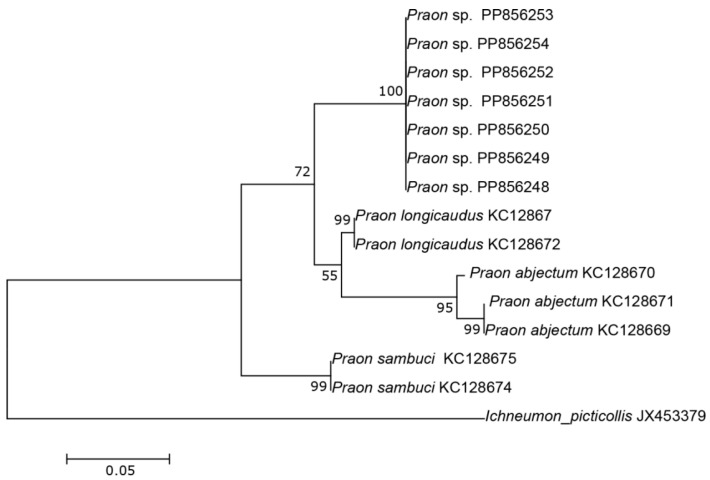
The evolutionary history of the *Praon abjectum* species group, inferred by using the Maximum Likelihood method based on the Tamura–Nei model.

**Figure 4 insects-15-00518-f004:**
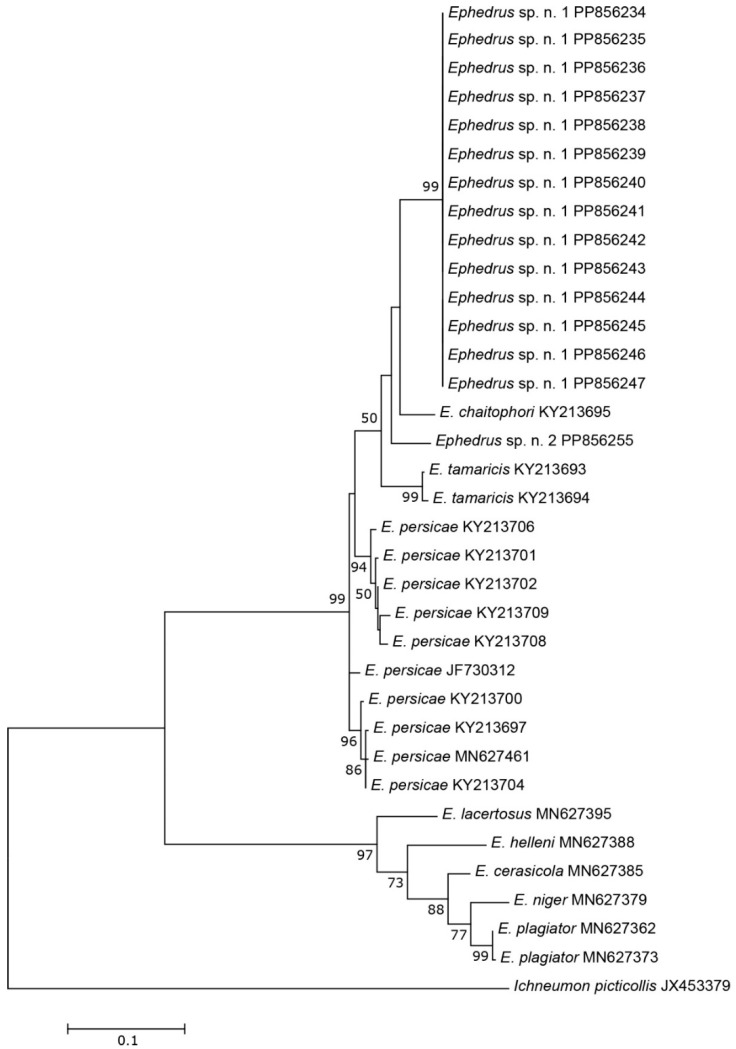
The evolutionary history of subgenus *Fovephedrus*, inferred by using the Maximum Likelihood method based on the Tamura–Nei model.

**Table 1 insects-15-00518-t001:** Malaise trap localities in Rogaland, Norway, with time periods, relevant to this study.

Locality	Sampling Period	GPS Coordinates
Sola, Forus, Røynebergsletta	29.06.2021–16.07.2021	58.90017 N, 5.68956 E
Sola, Forus, Røynebergsletta	15.06.2021–29.06.2021	58.90041 N, 5.68971 E
Svanholmen	15.08.2021–31.08.2021	58.88878 N, 5.71121 E
Suldal, Vårvik	27.06.2021–03.08.2021	59.54282 N, 6.64587 E
Sandnes, Forus	01.06.2022–15.06.2022	58.88880 N, 5.70937 E
Hå, Brusand	31.08.2020–17.09.2020	58.53837 N, 5.74777 E
Time, Linemyra	31.08.2021–30.09.2021	58.71556 N, 5.63867 E
Time, Linemyra	01.08.2021–30.08.2021	58.71556 N, 5.63867 E
Time, Linemyra	29.06.2021–20.07.2021	58.71556 N, 5.63867 E
Time, Linemyra	21.07.2021–31.07.2021	58.71556 N, 5.63867 E
Time, Linemyra	01.06.2021–29.06.2021	58.71556 N, 5.63867 E

## Data Availability

The new species sequences analysed in this study are deposited in the GenBank (https://www.ncbi.nlm.nih.gov/genbank/) under accession numbers PP856231–PP856255.

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
