# Peer review of "Uncovering Norway: Descriptions of Four New Aphidiinae Species (Hymenoptera, Braconidae) with Identification Key and Notes on Phylogenetic Relationships of the Subgenus Fovephedrus Chen"

_insects, 2024, doi:10.3390/insects15070518_

Round 1

Reviewer 1 Report

Comments and Suggestions for Authors

Fine addition to the fauna of Norway. See some minor proposed changes below. Figures are too dark.

Some minor changes are proposed:

Line 146 haired => setose (hairs are typical for mammals)

Line 165 rest of the body => remainder of body

Lines 182/227/281/337 = mesoscutum => = mesoscutum + scutellum [mesonotum is not same as mesoscutum]

Lines 183/269/282 sheaths => sheath (one sheath, two valves)

Lines 197/198 hairless => glabrous

Lines 264/265 hairs => setae

Line 354 members => taxa [members is too general]

Comments on the Quality of English Language

Fine, but see uploaded file

Author Response

Dear Editor and Reviewers,

Please find enclosed the revised version of our manuscript (ID insects-3069052) entitled: “Uncovering Norway: descriptions of four new Aphidiinae species (Hymenoptera, Braconidae) with identification key and notes on phylogenetic relationships of the subgenus Fovephed-rus Chen”.

We would like to thank you for very constructive comments and suggestions that will improve our manuscript. All text corrections are inserted into Word file using track changes. Our replies to the reviewer’s comments to the first version of the MS can be found below.

Korana Kocić

Detailed response to Reviewer 1

Comment 1.

Fine addition to the fauna of Norway. See some minor proposed changes below. Figures are too dark.

We changed brightness of the figures.

Comment 2.

Line 146 haired => setose (hairs are typical for mammals)

Changed.

Comment 3.

Line 165 rest of the body => remainder of body

Changed.

Comment 4.

Lines 182/227/281/337 = mesoscutum => = mesoscutum + scutellum [mesonotum is not same as mesoscutum]

Changed throughout the manuscript.

Comment 5.

Lines 183/269/282 sheaths => sheath (one sheath, two valves)

Changed throughout the manuscript.

Comment 6.

Lines 197/198 hairless => glabrous

Changed to glabrous.

Comment 7.

Lines 264/265 hairs => setae

Changed from hairs to setae throughout the manuscript.

Comment 8.

Line 354 members => taxa [members is too general]

Changed from members to taxa

Reviewer 2 Report

Comments and Suggestions for Authors

This is a worthwhile paper from an expert group describing four new species of Aphidiinae from Norway.  I recommend that it is accepted for publication subject to revision.

I think in general it is poor practice to describe species from a single specimen though given the molecular data it is probably just OK here.

I have two main recommendation that I strongly think should be implemented.

1. The quality of the English needs attention (see other section)

2. At least in the MS copy sent for review the illustrations are low quality and many of the key characters are difficult to see.  They need improvement.

I have also some other recommendation for the editors and authors to consider.

1.  There is some extraneous matter in the Introduction (much of the first paragraph could be removed) and some overlap with Discussion.  The low species count in Norway is obviously a sampling artefact and is emphasised too much.  These sections could be shortened.

2. For the new Aphidius and Praon I would like to see more information about existing identification resources and if keys exist for the relevant species group how the new species would come out.

3. The molecular tree for Ephedrus is very informative. Might it be possible to show equivalents for the other two species.

4. I recommend placing reference to the molecular data within the species description.

5.  I was unclear where the other Ephedrus sequence data came from and this should be explained in the Methods.

6. Has the BOLD database been examined for relevant sequences?

Comments on the Quality of English Language

I appreciate the difficulty of writing in a non-native language but the quality of the English is in many places not very good and it needs to be revised by a fluent speaker.  To give an example where it says "Praon breviantennalis sp.n. can easily be discriminated from another Praon .." I think you mean "... any other Praon ..." which means something different.

Author Response

Dear Editor and Reviewers,

Please find enclosed the revised version of our manuscript (ID insects-3069052) entitled: “Uncovering Norway: descriptions of four new Aphidiinae species (Hymenoptera, Braconidae) with identification key and notes on phylogenetic relationships of the subgenus Fovephed-rus Chen”.

We would like to thank you for very constructive comments and suggestions that will improve our manuscript. All text corrections are inserted into Word file using track changes. Our replies to the reviewer’s comments to the first version of the MS can be found below.

Korana Kocić

Comment 1.

“I think in general it is poor practice to describe species from a single specimen though given the molecular data it is probably just OK here.”

We agree with the reviewer that it is undesirable to provide species description based on a single specimen. Unfortunately, some species within subfamily Aphidiinae are very rare and just occasionally found. Despite the additional sample efforts, we were not able to collect more specimens. In the BOLD database, one more sequence of E. gardenforsi is present (also from Norway), but its status is private, therefore we were not able to contact collector or use the sequence in molecular analysis.

Comment 2.

“At least in the MS copy sent for review the illustrations are low quality and many of the key characters are difficult to see.  They need improvement.”

Figures in word file changed to higher resolution. We attached in the additional file all figures with high resolution.

Comment 3.

“There is some extraneous matter in the Introduction (much of the first paragraph could be removed) and some overlap with Discussion.  The low species count in Norway is obviously a sampling artefact and is emphasised too much.  These sections could be shortened. ”

We deleted overlapping text in Introduction and Discussion sections.

Comment 4.

“For the new Aphidius and Praon I would like to see more information about existing identification resources and if keys exist for the relevant species group how the new species would come out.”

We included three references in Discussion section about identification resources of the relevant species groups (Aphidius urticae s.str. and Praon abjectum species group).

Comment 5.

“The molecular tree for Ephedrus is very informative. Might it be possible to show equivalents for the other two species.”

We included phylogenetic trees for Praon breviantennalis and Aphidius norvegicus. The phylogenetic reconstruction was analysed within Praon abjectum species group and Aphidius urticae sensu stricto group. In our opinion both species are morphologically closest to these species groups.

Comment 6.

“I recommend placing reference to the molecular data within the species description.

GenBank accession numbers are added within species descriptions.

Comment 7.

“I was unclear where the other Ephedrus sequence data came from and this should be explained in the Methods.

We added in Methods section sentences, explaining that 13 sequences of subgenus Fovephedrus were acquired from GenBank, as well as six sequences of subgenus Ephedrus. Since we now added phylogenetic trees for Aphidius and Praon, we also added their sequence acquisition in this section.

Comment 8.

Has the BOLD database been examined for relevant sequences?

Yes, both GenBank and BOLD database were examined. In the BOLD database we discovered 3 additional sequences of Aphidius norvegicus, all from Norway, deposited in the Natural History Museum in Oslo. We were able to inspect the specimens via photos provided by the curators and confirm it is the same species.